# Texture-based classification of confocal laser endomicroscopy images for Barrett's esophagus surveillance

Giacomo Nardi[1], Marzieh Kohandani Tafreshi[1], Jessie Mahé[1], Nicholas Ayache[2], and François Lacombe[1]

[1] Mauna Kea Technologies, 9 Rue d'Enghien, 75010 Paris, France
{giacomo, marzieh, jessie, francois}@maunakeatech.com
[2] INRIA Sophia Antipolis - Méditerranée 2004 route des lucioles - BP 93 06902
Sophia Antipolis Cedex France
Nicholas.Ayache@inria.fr

**Abstract.** Barrett's esophagus is a complication of gastroesophageal reflux diseases that generates a transformation of esophagus epithelium turning into adenocarcinoma with a high risk. The surveillance of the changes in the esophageal mucosa is primordial to estimate the cancer progression. Confocal laser endomicroscopy is a novel imaging technique allowing physicians to perform *in-vivo* and *real-time* histological analysis in order to decrease the number of biopsies needed for the diagnosis. This paper uses the notion of local density function to extract characteristic morphologies of tissues. This allows us to define a novel classification method for Barrett's images based on fractal textures. The method performs particularly well on pre-cancer stages with an overall accuracy of 89.2%.

**Keywords:** Barrett's esophagus · confocal laser endomicroscopy · texture analysis · fractal local density · image classification.

## 1 Introduction

### 1.1 Medical Context

Barrett's esophagus designates a transformation (metaplasia) of the tissue lining the inside of the esophagus into intestinal or gastric-type tissue [3]. The main cause of Barrett's esophagus is the gastroesophageal reflux that induces such a modification in order to make the tissue more resistant to acid exposure. Intestinal and gastric metaplasia are highly associated to further transition into esophageal adenocarcinoma, so that they demand a periodic follow-up.

The surveillance of Barrett's esophagus evolution is traditionally obtained through microscopic analysis of several biopsies taken during endoscopies. Of course, such a process prevents from a complete examination of the tissue and is highly invasive for the patient. Moreover, biopsy results are not immediately available. Concerning these drawbacks, a new and promising technology is given

by Cellvizio, developed by Mauna Kea Technologies, Paris. This endomicroscopy system enables the practitioners to perform in-vivo confocal microscopy (*optical biopsy*) and provides them instantaneously with microscopic images in a minimally-invasive manner.

This technology is beneficial on several levels. Of course, to have an easier follow-up in case of cancer, but it also enables a closer analysis for earlier stages. In fact, when there is no visible sign of metaplasia or cancer, a few biopsies are taken by physicians and the pre-cancer stages are often no detected.

Classification of microscopic Barrett's esophagus images is then a crucial challenge to facilitate physician's decision-making in both pre-cancer and cancer stages.

## 1.2   Previous works

The standard classification for Barrett's esophagus is made through four classes that correspond to the main stages of the disease (see Fig. 1): Squamous Epithelium (SE), Intestinal Metaplasia (IM), Gastric Metaplasia (GM), Dysplasia or Cancer (DC).

These stages clearly show the transformation of the healthy epithelium (SE), recognizable by its tile-like appearance, into the cancerous tissue (DC) characterized by a disorganization at the cellular layer.

Intestinal and gastric metaplasia are respectively characterized by the appearance of both columnar mucosa and globet cells (IM) and gastric pits (GM).

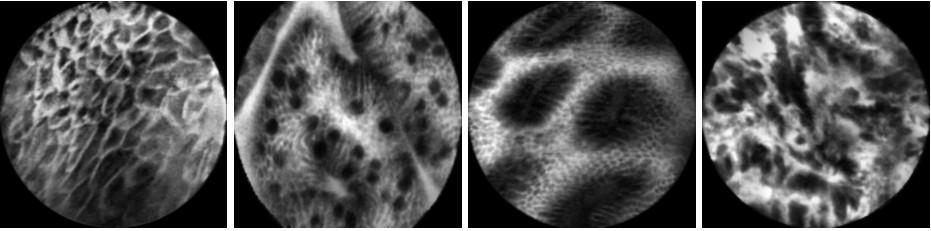

**Fig. 1.** From left to right : squamous epithelium (SE), intestinal metaplasia (IM), gastric metaplasia (GM), dysplasia or cancer (DC)

In [12], a binary tree classifier is defined to distinguish IM, GM, and DC on the basis of LBP-textures and Level-Sets geometric information of confocal images (overall accuracy of 96%). In [5], the proposed classification model improves images via an *ad-hoc* filter to extract several features (GLCM, LBP, fractal textures, wavelet features) achieving an overall accuracy of 90.4% for classifying the IM class. In [4], a fractal-based filter is used to improve images before extracting features (LBP, GLCM, fractal features) with an overall accuracy of 96% over the four previous classes. Similar techniques are used in [6, 7]. A deep learning

model using a convolutional neural network is proposed in [9] to distinguish IM, GM, and DC (overall accuracy of 80.7% on a small dataset).

### 1.3   Contributions

The goal of this work is to define a suitable texture-based algorithm for Barrett's images acquired by the Cellvizio system. This device has a high frame rate (around ten images per second), but the quality of images is quite low (SNR low, frequent local changes of illumination, non-rigid distortions due to the contact of the probe with the tissue).

Our first contribution consists in using the notion of local density function for image preprocessing. This allows us to detect cellular structures in case of changes of illumination so that a better segmentation is possible.

The second contribution consists in defining a novel texture based on the fractal properties for the level sets of density functions. To our knowledge this is the first paper using these features for endomicroscopic images. An accuracy of 89.2% is obtained over all the pre-cancer stages.

Finally, in comparison to the previously cited studies about Barrett's images classification, our result is obtained on a large dataset which proves the robustness of texture-based classification for pre-cancer stages. Moreover, these studies use a different dataset collecting images acquired with a different technology. For this reason, we can not consider this results as a reference, and a comparison with different methods is given.

## 2   Data and methods

### 2.1   Dataset

The dataset consists of 1694 images, collected from 31 patients throughout several clinical trials : 362 images of SE (12 patients), 560 images of IM (9 patients), and 772 images of GM (11 patients).

All images have been acquired by the Cellvizio system and are available on the website *www.cellvizio.net*. The acquisition is made by a confocal mini-probe GastroFlex UHD (around 30000 optical fibers, depth of observation of 60 $\mu m$, field of view of 240 $\mu m$, and a resolution of 1 $\mu m$) with a frame rate of around ten images per second.

Cellvizio is a fiber-bundle endoscopy system producing circular images [2]. In the following, in order to keep the maximum information of each image and avoid border effects, the largest square inscribed in the circular bundle is considered.

### 2.2   Fractal features

In order to describe the different topologies appearing in the images some level-sets-based features are proposed in [12]. However, in our images, due to low SNR and frequent changes of illumination, intensity level-sets do not enable image segmentation.

To overcome this problem, the *local density function* (LDF) is used to define textures [11]. This takes into account the local growth of the signal instead of its intensity. For each pixel $x$ the local density function is defined as:

$$\text{LDF}(x) \;=\; \lim_{r \to 0} \frac{\log \mu(\text{B}(x,\text{r}))}{\log \text{r}}$$

where $B(x,r)$ denotes the circular neighborhood of $x$ of radius r and $\mu(B(x,r))$ denotes the sum of the intensity values into $B(x,r)$. We compute the density by a linear fitting of $\log \mu(B(x,r))$ against $\log r$ for radius values ranging from 3 to 13 pixels.

In [13], a LDF-based feature is proposed and is proved to be invariant to local changes of illumination. This descriptor is used in [8] to classify endoscopic images of colonic polyps. In Fig. 2 some examples of segmentation via intensity-values and LDF are shown. We can observe that LDF-based segmentation provides more details and it is invariant to variations of illumination.

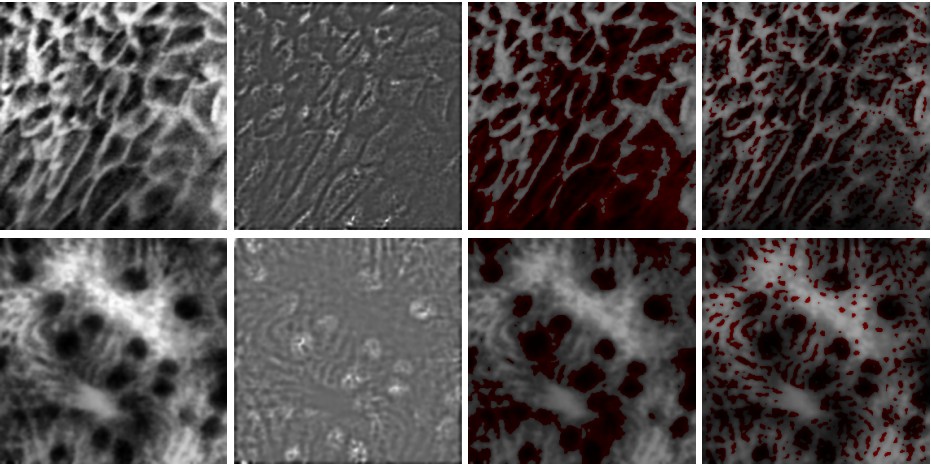

**Fig. 2.** From left to right : original image, local density function, intensity level set (in red), density level set (in red) for SE (top) and IM (bottom)

In the following, we consider the levels sets of local density functions corresponding to values ranging from 1.4 to 2.7 (these values have been set empirically after computation of all LDF-maps). For each level set we compute several fractal features : the ration of area to perimeter, fractal dimension of the level lines (computed via the box counting method), and the lacunarity describing how level sets fill the space (computed via the gliding box counting [1]). This finally leads to a 42-dimensional vector of fractal features.

### 2.3  LBP-texture features

A very common type of textures are Local Binary Patterns (LBP) [10]. For each pixel a binary pattern is defined by comparing its intensity with the intensity of pixels on its circular neighborhood. We use in particular rotation invariant uniform patterns (a uniform pattern has at most two 0 to 1 or 1 to 0 transitions). A multi-scale analysis is possible by choosing different radius for circular neighborhoods.

In the following, we consider radii = [3, 7, 11, 15, 19] and 16 neighbors for each circle. The choice of different radius enables a multi-scale analysis. Once LBP image has been computed, some statistical indicators of its histogram are considered: mean, variance, entropy, skewness, kurtosis. This finally leads to a vector of 25 LBP-features.

## 3   Results

In this section we present the results of our model and its comparison with other texture-based methods. We consider the classification of the pre-cancer stages (SE, IM, GM) and we discuss the difficulties to generalize our model to the four-classes problem (SE, IM, GM, DC).

### 3.1  Classification of pre-cancer stages

Experiments are carried out on a dataset of 1694 images of SE, IM, and GM. We consider 25 LBP-features and the 42-dimensional vector of fractal profiles.

We remind that, as the Cellvizio system has a high frame rate, then videos from the same patient may contain images highly correlated. For this reason we use the Leave-One-**Patient**-Out-Cross-Validation strategy (LOPO-CV) to validate our model. We also point out that this kind of validation is the most coherent with the algorithm application during medical procedures.

Moreover, we also evaluate the model via a splitting into training and test sets with no common patient (17 patients for the training set and 14 patients for the test-one).

Classification is made via both linear SVM and Random Forest (RF) classifier. All parameters (margin of error C for SVM and number of trees for Random Forest) are tuned by LOPO-CV on training sets.

We finally obtain an overall accuracy of 88.5% via the SVM classifier and of 89.2% via the Random Forest classifier by using the LOPO-CV strategy. The confusion matrices are shown in Tab. 1.

As pointed out in Section 1.3, the published papers on Barrett's classification cannot be used as a reference because of the different technology used to acquire images. We compare our model (LBP+FRACT) with the following methods : only LBP-textures defined in Section 2.3 (LBP), only fractal textures as defined in Section 2.2 (FRACT), and the Smart Atlas method [2].

**Table 1.** Confusion Matrix using LOPO-CV for pre-cancerous stages.

|  | SVM | | | Random Forest | | |
|---|---|---|---|---|---|---|
| **GROUND TRUTH** | **SE** | **IM** | **GM** | **SE** | **IM** | **GM** |
| **SE** | 328 | 17 | 17 | 315 | 28 | 19 |
| **IM** | 15 | 511 | 34 | 19 | 509 | 32 |
| **GM** | 41 | 71 | 660 | 50 | 35 | 687 |

The Smart Atlas method performs image classification combining a content-based image retrieval (CBIR) [2] with a k-nearest neighbors (k-NN) voting scheme.

Tab. 2 summarizes the comparison results showing that our algorithm outperforms the other methods. We also point out the high accuracy of FRACT confirming that fractal textures based on local densities well characterize the tissue morphologies.

**Table 2.** Accuracy comparison for the pre-cancer stages via LOPO-CV and splitting

|  | ACCURACY (LOPO-CV) | | ACCURACY (SPLIT) | |
|---|---|---|---|---|
| **METHOD** | **SVM** | **RF** | **SVM** | **RF** |
| **Smart Atlas** | 65.5% | | 65.7% | |
| **LBP** | 69.1% | 68.7% | 80.1% | 67% |
| **FRACT** | 84.9% | 83.2% | **90**% | 80.4% |
| **LBP+FRACT** | **88.5**% | **89.2**% | **89**% | **86.9**% |

Fig. 3 shows some examples of correct classifications for SE, IM and GM classes highlighting the variability of tissue morphologies within each class.

### 3.2   Discussion

The previously defined model is based on texture-features which describe the cellular organization. This enables our classifier to well distinguish pre-cancer stages that have well defined tissue architectures.

The case of cancerous tissues is more complex. In fact, as explained above, cancer stage is characterized by a highly disorganized cellular architecture, so that the presented characterization by morphologies segmentation is less adapted.

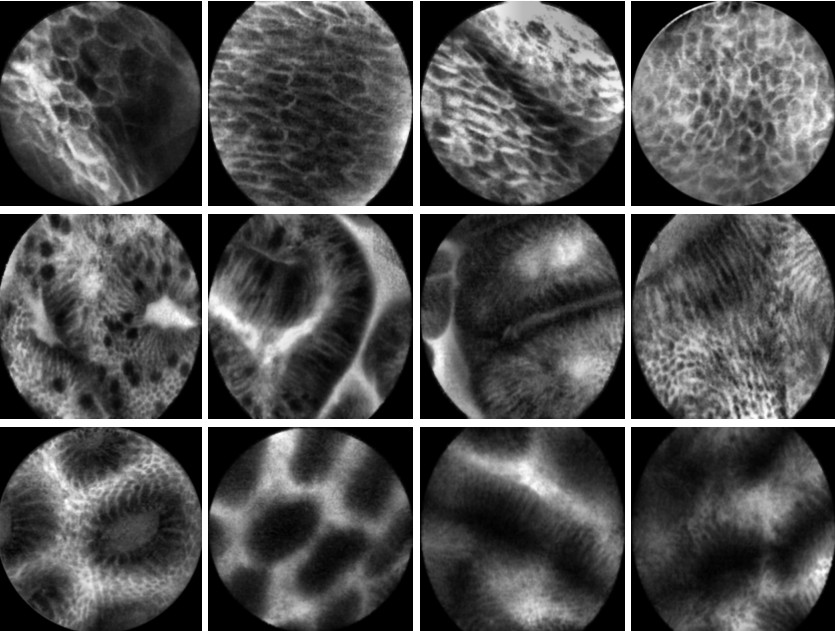

**Fig. 3.** Examples of correct classifications for SE (top), IM (middle) and GM (bottom)

Next step consists in defining new features characterizing the different grades of cancer. This is actually a challenging problem because of the architecture variability in cancerous tissues. Moreover, no criterion exists yet to distinguish the different grades of cancer. Then, a finer analysis on a larger dataset is needed, in order to define suitable textures based on a collection of morphologies within the cancer class.

## 4   Conclusion

This paper introduces a method to extract characteristic structural features from microendoscopic images with low SNR and unstable intensity. This provides a more robust description of the cellular architecture of tissues compared to intensity-based segmentation.

We used this method to define a classifier of pre-cancer stages for Barrett's esophagus. Previous results show that LBP and fractal features well characterize SE, IM, and GM stages. A classification result shows an overall accuracy of 89.2% on a large dataset.

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
