# OpenReview forum: "Texture-based classification of confocal laser endomicroscopy images for Barrett’s esophagus surveillance"
_MICCAI.org/2019/Workshop/COMPAY — Submitted to COMPAY 2019_

### Official Review · AnonReviewer3 · 2019-08-09
**Interesting in vivo real-time histological data collected by confocal endomicroscopy, classified by fractal features. Several gaps in the methodology description as well as poor evaluation in relation to other methods makes the study less interesting.**

**Rating:** 5
**Confidence:** 5

**Review:**

Concerns:
The authors confuse classification and segmentation, where the paper is about image classification while references focus on segmentation (which is a different problem and would need a different type of ground truth and evaluation).

The authors make an incomplete comparison to previous methods; presenting the results of previous methods applied to a different dataset is pointless. If they want to compare to them then the same dataset should be used. Robustness cannot be claimed if there is no comparison.

The strong emphasis of the company providing the data makes me feel there may be a conflict of interest (especially as all but one of the authors are associated with the same company). The paper should focus more on the details of the proposed method. Data as such provides a unique set of possibilities due to its time dimension, and future studies making use of these aspects could be interesting.



The LDF is central as it is the basis for the later extraction of fractal features. However, many things are unclear:
•	The LDF is defined as a limit as r tends to 0 but r is still only in the range 3 to 13. Such a limit would make sense in a continued valued function but not in a discrete function (image).
•	It is not clear how the LDF is invariant to changes in illumination. From the example images it seems that a single location of tissue with uneven illumination would not produce the same LDF.
•	How is the final LDF image computed? The "best" of the different radii? a sum? a MIP?

After applying the LDF, levelsets are used. Things are again not clear:
•	There are infinite level sets, even between 1.4 and 2.7, which and how many level sets are taken? Is it the same thing as a threshold here, and how was this threshold selected? The red color in the images is hardly visible; showing the result as a binary image would improve visibility.
•	How is the perimeter of the levelset calculated? What is the 42D vector of fractal features composed of? Do all fractal features make sense when resolution is limited? How do fractal features correlate between consecutive time steps, showing more or less the same region?

LBPs are not introduced and the section 2.3 appears as if floating for no reason only to be mentioned later.

An initial summary of the method should be introduced at the very beginning of section 2 and not as single pieces that don't seem to be connected.

SVM and RF can be trained in many ways and the way the authors used is not specified.

If the papers used on similar images can't be used as reference then the percentage presented for the LBP+FRACT can't be compared to any of the methods presented in those previous papers. The only valid comparisons are between LBP+FRACT,LBP,FRACT.

In the discussion, the model is not really defined, there are components mentioned and their connections is loosely described.

It seems rushed to discard deep learning, particularly if textures are considered the best way to classify these images. Different architectures can provide different results and only one paper was mentioned. There are papers combining CNN+LBP, we suggest: "Towards Automated Multiscale Imaging and Analysis in TEM: Glomerulus Detection by Fusion of CNN and LBP Maps" 10.1007/978-3-030-11024-6_36

An initial quality control step is mentioned, but no details for this process is available. Was it visual?

Minor comments:
In section 2.2, the word topologies is wrongly used. Probably meaning morphology.

---

### Official Review · AnonReviewer4 · 2019-08-15

**Rating:** 5
**Confidence:** 4

**Review:**

SUMMARY
The authors present a method classifying endomicroscopy images of esophagus to squamous epithelium, intestinal metaplasia, gastric metaplasia pre-cancer stages. To do so, they extract features from various level sets of the local density function and from local binary patterns.

My biggest concern is that it is hard to put the performance of the algorithm in context. The authors cite many other algorithms that work on similar images where tey get the inspiration, yet they only implement a different, "traditional" method. One of the cited algorithm is from a challenge on the same task, where the dataset of the challenge is still available. For fair comparison, the authors need to either run their proposed method on the open access dataset, or implement some of the published methods and apply it on their data set.

REMARKS:
1. The references should be in order.

2. There is too much advertisement for Cellvizio. I understand that the authors are from the company that produces the equipment, but several features are completely irrelevant for the manuscript. Anyone who is interested in the product can find it on the internet, there is no need to even force a url in the manuscript.

3. The dataset of [9] is still available. Since the authors compare their proposed method against a single traditional method. It would be informative to see how their algorithm performs on an open-access dataset for comparison with other methods.

4. Why is it not possible to compare it against [12]?

5. The composition of the dataset is not clear. The "frame rate" of the device is mentioned repeatedly. So the images are actually highly correlated frames of videos? If so, how many videos, or image sequences are there in the data set?

6. Since the authors mention the frame rate of the device, it would be interesting to know the speed of the classification on a single frame? Is it real time?

7. What are the size of the actual processed images? It is mentioned that the largest fitting rectangle in the circular field of view, but how many pixels in the end?

8. "DF-based segmentation provides more details and it is invariant to variations of illumination" Segmentation of what? What do the authors mean by more details?

9. For reproducibility reasons it would be useful to know the actual values between 1.4 to 2.7 that were used for calculating the levels sets.

10. The authors mention no details, or values of the parameters of the SVM or the Random Forest classifier.